# Resveratrol-Induced White Adipose Tissue Browning in Obese Mice by Remodeling Fecal Microbiota

**DOI:** 10.3390/molecules23123356

**Published:** 2018-12-18

**Authors:** Weiyao Liao, Xiaohan Yin, Qingrong Li, Hongmin Zhang, Zihui Liu, Xinjie Zheng, Lin Zheng, Xiang Feng

**Affiliations:** 1Department of Nutrition, School of Public Health, Sun Yat-sen University; Guangzhou 510080, China; liaowy6@mail2.sysu.edu.cn (W.L.); yinxh5@mail2.sysu.edu.cn (X.Y.); liqr9@mail2.sysu.edu.cn (Q.L.); zhanghm7@mail2.sysu.edu.cn (H.Z.); liuzih@mail2.sysu.edu.cn (Z.L.); zhengxj27@mail2.sysu.edu.cn (X.Z.); zhenglin@mail.sysu.edu.cn (L.Z.); 2Guangdong Provincial Key Laboratory of Food, Nutrition and Health, Guangzhou 510080, China

**Keywords:** resveratrol, gut microbiota, WAT browning, Sirt1

## Abstract

Promoting the browning of white fat may be a potential means of combating obesity. Therefore, in this study, we investigated the effect of resveratrol (RES) on the body weight and browning of white fat in high-fat diet (HFD)-induced obese mice and the potential associated mechanism in vivo. Eight-week-old male mice were randomized to receive different treatments: (1), chow without any additional treatment (chow); (2), chow plus 0.4% resveratrol (chow-RES); (3), HFD without any additional treatment (HFD); and (4), HFD plus 0.4% resveratrol (HFD-RES). After 4 weeks of feeding, additional 8-week-old male recipient mice were randomly allocated to the following 4 treatments: (5), HFD and received feces from chow-fed mice; (6), HFD and received feces from chow-RES-fed mice; (7), HFD and received feces from HFD-fed mice; and (8), HFD and received feces from HFD-RES-fed mice. RES treatment significantly inhibited increases in fat accumulation, promoted the browning of white adipose tissue (WAT) and alleviated gut microbiota dysbiosis in HFD-fed mice. Subsequent analyses showed that the gut microbiota remodeling induced by resveratrol had a positive role in WAT browning, and sirtuin-1 (Sirt1) signaling appears to be a key component of this process. Overall, the results show that RES may serve as a potential intervention to reduce obesity by alleviating dysbiosis of the gut microbiota.

## 1. Introduction

Obesity is a severe health problem due to its associations with various adverse health consequences, and the prevalence of obesity has continuously increased worldwide [1]. The characteristics of obesity are immoderate accumulation and storage of fat in the body. The best way to treat obesity is to promote a decrease in energy intake and an increase in energy expenditure [2]. Thus, the current method of weight loss is calorie restriction and exercise. However, because this method is difficult to implement and the lost weight is easy to regain, new methods of treating obesity need to be explored [3].

In the search for new obesity treatments, brown adipose tissue (BAT) has gained increasing attention. BAT is primarily distributed around the neck and scapular region of the body, and its function is to decompose and oxidize fatty acids synthesized by triglycerides and to consume calories through non-shivering thermogenesis. Brown fat adipocytes contain multiple small fat droplets and more mitochondria than white adipose tissue (WAT) [4]. Uncoupling protein 1 (UCP1), a key protein in the non-shivering thermogenesis process, is located in the mitochondria. BAT is primarily present in infants and children, whereas most adipose tissue in adults is WAT, with only a small amount of BAT. Interestingly, a process called browning can transform WAT into beige adipose tissue. Although beige adipocytes have different differentiation pathways and gene expression patterns compared to brown adipocytes, they have the similar physiological role in the body of having a high metabolic rate and generating heat. Interestingly, studies have shown that inducing WAT browning can prevent obesity caused by diet and improve insulin sensitivity [5,6,7].

In recent years, the gut microbiota has been identified as a factor in the development of obesity. In 2004, Backhed first proposed that gut microbiota may affect the synthesis and metabolism of lipids, believing that obesity is related to the community structure of the gut microbiota. Gut communities with a high abundance of Firmicutes and a low abundance of Bacteroides can extract and absorb the calories from food efficiently and lead to obesity [8,9]. When mammals are exposed to cold temperatures, changes such as increased BAT weight and WAT browning occur in the body. These changes were shown to be likely promoted by intestinal microorganisms after gut microbiota transplantation studies [10].

Resveratrol (RES; 3,5,4′-trihydroxytrans-stilbene) is a nonflavonoid polyphenol compound containing the structure of astragalus, which is naturally found in a wide variety of plants, such as grapes, pines, knotweed and peanuts. Besides its multiple properties, including antioxidant, anti-inflammatory, anti-carcinogenic and anti-adipogenic effects, there are also some novel properties that have been found recently. It was reported that RES is involved in cardioprotective, neuroprotective, anticancer and antiviral activities [11,12,13,14,15]. The results of previous studies have shown that resveratrol activates sirtuin-1 (Sirt1) and enhances mitochondrial respiration, which in turn increases energy expenditure [16]. Sirt1 is involved in many metabolic activities, including modulating hepatic lipid and systemic glucose homeostasis and remodeling adipose tissues by regulating the activity of metabolic regulators, such as peroxisome proliferator-activated receptor gamma (PPAR-γ) and the cofactor, peroxisome proliferator-activated receptor gamma coactivator-1 alpha (PGC-1α) [17]. In previous studies, RES was observed to potentially alter the composition of the gut microbiota, to ameliorate the increases in weight of the body and the adipose tissues, and to improve glucose homeostasis in HFD-fed mice [18,19]. In this study, we further examined the effect of resveratrol on body weight and the browning of white fat in HFD-fed mice and the underlying mechanism in vivo. The aim of this study was to assess whether RES-induced browning of white fat is partly mediated by gut microbiota remodeling.

## 2. Results

### 2.1. RES Mitigated HFD-Induced Obesity in Mice

We first assessed the impact of RES on body weight gain and the distribution of white adipose in chow-fed and HFD-fed groups of mice. The results showed remarkable increases in body weight gain, perigonadal visceral adipose tissue (pgVAT) and inguinal adipose tissue (ingSAT) accumulation in the HFD-fed mice (Figure 1A–C). However, these increases were significantly inhibited in HFD-fed mice that received the RES treatment (Figure 1A–C). Morphologically, the size of adipocytes was decreased in WAT in both of the RES treatment groups (Figure 1D), suggesting that RES has anti-obesity effects. No significant differences in the amounts of energy intake by the animals were observed among the different groups (Appendix A), suggesting that the anti-obesity effect of RES was not due to reduced energy intake.

### 2.2. RES Modulated Glucose Homeostasis in HFD-Fed Mice

As mentioned above, RES can enhance mitochondrial respiration, which in turn improves insulin sensitivity. To determine whether RES affects glucose homeostasis, we performed an oral glucose tolerance test (OGTT) and intraperitoneal injection of insulin tolerance test (IPITT) at the end of the intervention period. Marked increases in the fasting blood glucose levels and in the areas under the curve (AUCs) were observed during OGTT and IPITT in the HFD-fed group. As expected, RES promoted glucose homeostasis in mice, with the RES-treatment group showing increases in fasting blood glucose levels and glucose intolerance (AUCs of the OGTT and the IPITT) compared with those observed in the HFD-fed mice (Figure 2A–D).

### 2.3. RES Reversed HFD-Induced Dysbiosis

Changes in dietary habits (particularly with respect to a HFD) can change the composition of the gut microbiota. We performed a 454-pyrosequencing analysis of the bacterial 16S rRNA gene (V4-V5 region) in stool samples. We observed a marked decrease in the bacterial alpha diversity indices in the RES-treated groups (*p* < 0.05) (Figure 3A). Notably, we observed that the RES treatment reduced the abundances of the Bacteroidetes and the Proteobacteria phyla and increased the abundance of the Firmicutes phylum in HFD-fed mice to levels similar to those observed in chow-fed mice (Figure 3B). Unweighted UniFrac-based PCoA showed that each treatment group had a distinct clustering of microbiota composition (Figure 3C). LEfSe analysis was performed to identify the specific bacterial biomarkers that were modulated by the HFD and RES interventions (Figure 3D). Microbial taxa enriched in the microbiota from mice fed the HFD, chow, HFD + RES or chow + RES diets were identified by the Linear Discriminant Analysis (LDA) score. Overall, these results show that RES reversed the gut dysbiosis observed in HFD-fed mice by modulating the gut microbiota composition to one similar to that of chow-fed mice.

### 2.4. RES Fecal Transplants Modulated Gut Microbiota Composition

We transplanted feces from mice fed chow (Chow→HFD), chow + 0.4%RES (CHOW.RES→HFD), HFD (HFD→HFD) or HFD + 0.4%RES (HFD.RES→HFD) and analyzed the gut microbiota of the recipient mice. The results indicated that the Firmicutes/Bacteroidetes ratio in the recipient mice (Figure 4B) was similar to the donor mice (Figures 3B and 4B). We observed a decrease in the bacterial alpha diversity in the gut microbiota of mice transplanted with feces from the chow + 0.4%RES mice (*p* < 0.05) (Figure 4A). The PC1 axis of the PCoA clearly separated the bacterial communities according to different treatments, and multivariate nonparametric ANOVA showed that PC1 explained approximately 19.23% of the variability in the microbiota composition (Figure 4C). LEfSe analysis was performed to identify the specific bacterial biomarkers that were modulated by fecal transplants and diet (Figure 4D).

### 2.5. RES Fecal Transplants Reduced HFD-Induced Obesity in Mice

To determine whether the anti-obesity activity of the gut microbiota of RES-treated mice can be transferred to recipient HFD-fed mice, we performed fecal transplants and examined the obesity-associated indices described above. Our results show that mice in the Chow→HFD, CHOW + RES→HFD, or HFD + RES→HFD groups exhibited reduced body weight, ingSAT and pgVAT fat weight compared with mice in the HFD→HFD group (Figure 5A–C). Furthermore, morphological changes in adipocytes were observed in mice that received fecal transfers from the Chow + RES and HFD + RES groups, as increased numbers of smaller adipocytes with a multilocular phenotype were observed in the WAT of mice from these groups (Figure 5D). Mice that received a fecal transfer from mice treated with RES exhibited reduced obesity and fat accumulation compared with mice that received a fecal transplantation from HFD-fed mice. A significant decrease in energy intake by the animals in the HFD→HFD group was observed (Appendix A), suggesting that the anti-obesity effects of RES were not due to reduced energy intake. These findings indicated that the WAT-browning effect of RES in HFD-fed mice may be mediated by gut microbiota.

### 2.6. RES Fecal Transplants Improved Glucose Homeostasis in HFD-Fed Mice

We also performed OGTT and IPITT in recipient mice to determine whether RES-remodeling of the gut microbiota would influence glucose homeostasis. In mice from the Chow→HFD, CHOW.RES→HFD or HFD.RES→HFD groups, a decrease in the fasting blood glucose levels and a decrease in the AUCs during OGTT and IPITT were observed (Figure 6A–D). As expected, fecal transplants from RES-treated mice promoted glucose homeostasis. These results suggest that RES-remodeling of the gut microbiota may help promote glucose homeostasis in HFD-fed mice.

### 2.7. RES and RES Fecal Transplants Modulated the Expression of Proteins Associated with Energy Expenditure and WAT Browning

Based on the above results, we further investigated the brown fat-specific markers in the ingSAT and pgVAT depots. Western blotting results showed that UCP1 and SIRT1 expression was apparently reduced in both the ingSAT and pgVAT of the HFD-fed mice compared with that of mice fed chow. In contrast, RES intervention markedly increased the expression of UCP1 and SIRT-1, which are involved in mediating the effect of thermogenesis and promoting WAT browning (Figure 7A–D). To investigate the mechanism underlying the effect of RES intervention on Sirt1 expression, we further examined the expression of the downstream proteins PGC-1α and PPAR-γ in ingSAT and pgVAT by Western blotting assay. The results indicate that PGC-1α and PPAR-γ protein expression was increased in mice administered RES compared with that in mice fed a HFD alone (Figure 7A–D), and recipient mice had similar expression levels of proteins in the Sirt1 protein pathway (Figure 7E–H). These results demonstrate that RES intervention can promote WAT browning by modulating the gut microbiota and suggests that these RES-mediated effects may be due to modulation of the Sirt1 protein pathway.

## 3. Discussion

Obesity is a serious worldwide health problem, and a HFD plays an important role in the development of visceral obesity in mammals [20]. The results of our previous study and a great deal of other evidence suggests that RES has anti-obesity effects, with the mechanisms underlying these effects having been reported to be downregulators of adipogenic and inflammatory processes and promotors of fat cell apoptosis [21,22,23]. However, a recent study showed that RES can promote the browning of white fat depots, which is believed to be a new treatment modality in the fight against obesity [24]. Chevalier demonstrated that increased browning partially protects against obesity and insulin resistance and that this effect is mediated by microbiota [10]. Because the systemic bioavailability of RES was previously shown to be very low but its concentration was very high in gut tissue [25,26], we hypothesize that RES may have a WAT-browning effect by targeting the gut microbiota, which in turn improves fat storage and metabolism. In addition, since RES is a key regulator of Sirt1 signaling, our study examined whether the Sirt1 pathway is involved in the underlying mechanism of gut microbiota-mediated host thermogenesis homeostasis.

Although the gut microbiota-mediated anti-obesity function of RES is consistent with previous studies [18,27], analyses were not performed to assess whether the alteration of the gut microbiota is essential for the RES-mediated improvement in thermogenesis homeostasis. Our results indicate that RES is sufficient to decrease fat content in both ingSAT and pgVAT and increase insulin sensitivity. We then further investigated if microbiota remodeling is the mechanism for the anti-obesity activity of RES.

The gut bacterial phyla Bacteroidetes and Firmicutes have been reported to affect energy metabolism homeostasis [28]. The results of previous studies suggested that a HFD can induce gut dysbiosis [29]—as indicated by observations of decreased and increased intestinal Bacteroidetes and Firmicutes abundances, respectively, as well as a higher abundance of endotoxin-bearing Proteobacteria. Audrey reported that the administration of wheat arabinoxylan, a type of prebiotic, can alter the abundances of Bacteroides and Firmicutes and reduce weight gain [30]. In the present study, the relative abundances of the gut-associated bacterial taxa showed a HFD-induced dysbiosis, and HFD-induced obese mice had lower Bacteroidetes/Firmicutes ratios and a higher abundance of Proteobacteria than chow-fed mice. However, RES supplementation significantly alleviated the gut microbiota dysbiosis induced by HFD, restoring the Bacteroidetes-to-Firmicutes ratio to that observed in chow-fed mice, and significantly inhibited the growth of Proteobacteria. We also observed an increase in the growth of the probiotics Lactobacillus and Bifidobacterium in mice, both of which were previously reported to reduce obesity [31,32,33]. This result suggests that RES may exert an anti-obesity activity by modifying the levels of specific bacterial phyla, such as Firmicutes, Bacteroidetes and Proteobacteria (Figure 4B). Donor mice and mice that received a fecal microbiota transfer (FMT) had similar but not identical gut microbiota compositions, which occurred because dietary changes or fecal transfers can modulate the gut microbiota (Figure 5B). However, mice that received a RES-FMT still had higher Firmicutes-to-Bacteroidetes ratios than mice that received a HFD-FMT.

Recently, a third type of adipocyte, called “beige” adipocytes, was discovered that exhibits a gene expression profile that is similar but not identical to that of classic brown adipocytes. [34]. Beige adipocytes are characterized by their multilocular lipid droplet morphology, high levels of mitochondria and expression of a set of brown-fat markers (for example, UCP1 and PGC-1α) [35,36]. Beige fat can be easily recruited and activated by β-adrenergic receptors or PPAR-γ [37], which can also activate mitochondrial respiration and energy expenditure in brown adipocytes [38]. Beige adipocytes in WAT are responsible for heat production and have beneficial effects with respect to anti-obesity, insulin resistance and hyperlipidemia [6,39]. RES-treated mice were observed to show increases in the expression of beige fat-related proteins and brown-like cells in WAT, and insulin sensitivity was also improved. These results suggest that RES could promote WAT browning in HFD mice. In order to determine whether this effect of RES-treated mice is caused by microbiota remodeling, we performed FMT in HFD mice. Interestingly, we observed similar improvements in RES-FMT mice. Our data appear to exclude potential direct effects of circulating resveratrol on target tissues and proved that the anti-obesity effect is transmissible via FMT from RES-treated mice to HFD-fed mice. These abilities of RES are transferrable through colonization, supporting the theory that browning of WAT is related to altering gut microbiota composition.

Although the results of a previous study showed that RES is a Sirt1 activator, the microbiota of RES-treated mice also activates Sirt1. A Sirt1 gain-of-function mutation induces BAT-like remodeling of white adipocytes in vivo and in vitro through PPAR-γ and PGC-1α deacetylation [40,41]. Moreover, Sirt1 also plays an important role in gut homeostasis [42,43]. First, Sirt1 sustains intestinal barrier function by enhancing crypt proliferation and suppressing villous apoptosis [44], fostering the expansion of intestinal stem cells in the gut [45], and by regulating tight junction-associated proteins during hypoxia, such as zonulin occludin-1, occludin and claudin-1 [46]. In HFD-fed animals, intestinal dysbiosis may allow LPS from gram-negative bacteria to leak from the intestinal lumen and enter the hepatic circulation, which in turn causes systemic inflammation in HFD-fed mice through activation of TLR4 signaling [47]. Macrophages present within adipose tissues were shown to be involved in innate immunity and can be activated as an M1 or M2 phenotype in response to stimuli. Polarized M1 macrophages are associated with inflammation, whereas polarized M2 macrophages promote WAT browning/beiging [48,49]. However, gut microbiota can regulate secondary bile acid metabolism, and secondary bile acid metabolism may directly modify Sirt1 and modulate the activity of transcription factors, such as farnesoid X receptor (FXR) and G-coupled membrane protein 5 (TGR5), to alter mitochondrial biogenesis. FXR is also a target of the NAD-dependent protein deacetylase Sirt1 [50,51,52,53,54], the role of which in microbiota-host interactions is beginning to be elucidated, shedding light on the role that gut microbiota-WAT browning crosstalk plays in energy metabolism [54,55].

## 4. Materials and Methods

### 4.1. Animals

All animal experiments were conducted in accordance with the guidelines of the Animal Ethics Committee of Sun Yat-sen University (No. 2017-008). Male C57BL/6J mice were purchased from the Guangdong Medical Laboratory Animal Center (Guangzhou, Guangdong, China) and were maintained under standard specific pathogen-free (SPF) conditions with free access to food and water. Mice were fed either a standard chow diet (10% of energy from fat; D12450b; Research Diet, USA) or a HFD (60% of energy from fat; D12492; Research Diet, New Brunswick, NJ, USA). Eight-week-old male mice were randomly allocated to the following 8 groups containing seven or eight animals each: (1) chow without any additional treatment (chow); (2) chow plus 0.4% resveratrol (chow-RES); (3) HFD without any additional treatment (HFD); (4) HFD plus 0.4% resveratrol (HFD-RES); (5) HFD and received feces from chow-fed mice (Chow→HFD); (6) HFD and received feces from chow-RES-fed mice (CHOW + RES→HFD); (7) HFD and received feces from HFD-fed mice (HFD + RES→HFD); and (8) HFD and received feces from HFD-RES-fed mice (HFD→HFD). The resveratrol-treated mice received 400 mg per kg dietary resveratrol (Sigma-Aldrich, St. Louis, MI, USA) that was mixed to homogeneity during the preparation of the diets.

### 4.2. Fecal Microbiota Transplantation (FMT)

Prior to microbial transplantation, recipient mice were treated with a 200 mL antibiotic cocktail (containing 1 g/L of ampicillin and metronidazole and 0.5 g/L of vancomycin and neomycin; Sigma-Aldrich, St. Louis, MI, USA) administered by oral gavage once a day for 3 days [56]. Fecal transplantations were performed as described in a previous study [57]. Eight-week-old male recipient mice (*n* = 7/group) were inoculated daily with 100 μL of the fresh transplant solution by oral gavage for 8 weeks before being sacrificed for subsequent analysis (Appendix A).

### 4.3. Antibodies

Antibodies against UCP1 (uncoupling protein 1; 1:1000; #14670), Sirt1 (sirtuin-1; 1:1000; #3931), PPAR-γ (peroxisome proliferator-activated receptor gamma; 1:1000; #2435), PGC-1α (peroxisome proliferator-activated receptor gamma coactivator-1 alpha; 1:1000; #2178) and a secondary antibody (anti-rabbit IgG, HtL; 1:5000) were purchased from Cell Signaling Technology (Danvers, MA, USA). An antibody against β-actin (1:20,000; NB600-501) and an anti-mouse IgG secondary antibody (1:20,000; sc-2005) were purchased from Santa Cruz Biotechnology (Dallas, TE, USA).

### 4.4. OGTT and IPITT

An oral glucose tolerance test (OGTT) and an intraperitoneal injection of insulin tolerance test (IPITT) was performed for each mouse at the end of the treatment period. Glucose tolerance tests were performed by oral gavage of glucose (2 g/kg) after 6 h of fasting, and ITTs were performed by intraperitoneal injection (0.75 U/kg) after a 5-h daytime fast. Blood glucose was measured with a glucose meter (Sannuo, Shenzhen, China) using blood collected from the tip of the tail vein.

### 4.5. Western Blotting

Adipose tissue proteins were extracted using a commercial RIPA lysis buffer (Beyotime, Shanghai, China) supplemented with 1% PMSF. Total protein lysates were separated on an 8–10% SDS-PAGE gel and then transferred to appropriate polyvinylidene fluoride (PVDF) membranes. The PVDF membranes were blocked with 5% nonfat milk for 1–2 h at room temperature and then incubated with the primary antibodies overnight at 4 °C. After being incubated with an HRP-conjugated secondary antibody, protein bands were developed using an ECL Reagent Kit (Thermo Fisher Scientific, Waltham, MA, USA). The primary and secondary antibodies are described in the Antibodies section above.

### 4.6. Gut Microbiota Analysis

DNA was extracted from each sample using a DNA extraction kit (Tiangen Biotech, Beijing, China). The V4-V5 regions of the bacterial 16S rRNA gene was amplified using the specific primers 515F and 907R with a 12 bp barcode. Primers were synthesized by Invitrogen (Invitrogen, Carlsbad, CA, USA). Samples with bright primary bands between 400–450 bp were used for further experiments. PCR products were mixed at equivalent ratios using GeneTools (Version 4.03.05.0, SynGene, Bengaluru, India), after which the PCR product mixtures were purified using an EZNA Gel Extraction Kit (Omega, La Chaux-de-Fonds, Switzerland). The appropriate primers were selected for amplification of each product, and in cases when the final primer sequence was not known, it was viewed in the mapping file of the analysis result package.

Sequencing libraries were generated using an NEBNext^®^ Ultra™ DNA Library Prep Kit for Illumina^®^ (New England Biolabs, MA, USA) following the manufacturer’s recommendations with index codes added. The library quality was assessed using a Qubit@ 2.0 Fluorometer (Thermo Fisher Scientific, MA, USA) and an Agilent Bioanalyzer 2100 system (Agilent Technologies, Waldbron, Germany). Finally, the library was sequenced on an Illumina HiSeq 2500 platform, and 250 bp paired-end reads were generated.

Sequences with ≥97% similarity were assigned to the same operational taxonomic unit (OTU), each of which represents a potential species. Alpha diversity was calculated using R (V2.15.3, Vienna, Austria) and data were visualized using R, and beta diversity and sample clustering analyses were performed using R.

### 4.7. Statistical Analysis

The results are presented as individual data points or as the means ± the standard error of the mean (SEM). Data were analyzed using SPSS 20.0 (SPSS, Chicago, IL, USA). Graphs were generated using Graph Pad Prism 7.0 (Graph Pad Prism, San Diego, CA, USA). In experiments comparing multiple groups, one-way analysis of variance (ANOVA) was used. OGTTs and IPITTs were analyzed using repeated measure two-way ANOVA with both time and group as sources of variation. *p*-values < 0.05 were regarded as indicating significance.

## 5. Conclusions

The results of this study demonstrate that RES treatment significantly inhibited fat accumulation, promoted browning of white adipose tissue (WAT) and improved gut microbiota dysbiosis in HFD-fed mice. Based on these improvements in RES-FMT mice, our data appeared to exclude direct effects of circulating resveratrol on target tissues and proved that the anti-obesity effect is transmissible via FMT from RES-treated mice to HFD-fed mice. Changes in the gut microbiota caused by intervention with RES induces WAT browning in HFD-fed mice (Figure 8), but the specific gut microbes that have the greatest impact on this process remain unknown. Although the specific mechanisms by which the RES-associated gut microbiota exerts its anti-obesity-promoting activity remain to be identified, Sirt1 signaling appears to be a key pathway associated with this process in diet-induced obese mice.

## Figures and Tables

**Figure 1 molecules-23-03356-f001:**
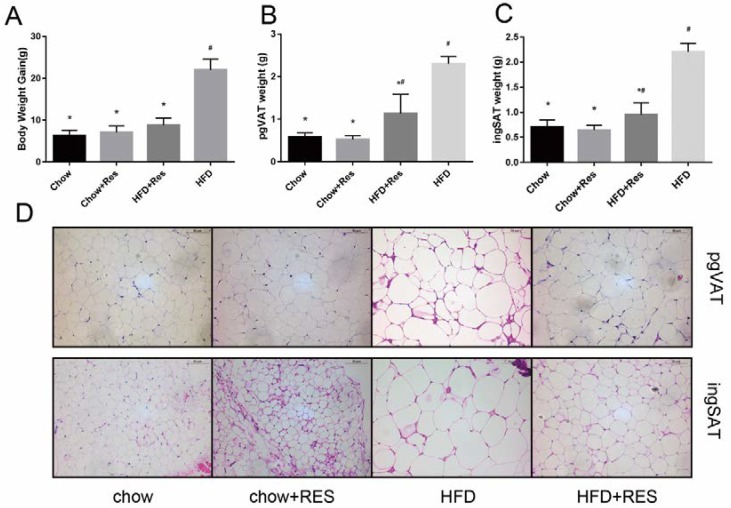
Resveratrol (RES) mitigated high-fat diet (HFD)-induced obesity in mice. Effects of RES treatment on (**A**) body weight gain (**B**) perigonadal visceral adipose tissue (pgVAT) weight (**C**) inguinal adipose tissue (ingSAT) weight and (**D**) perigonadal visceral adipocyte size and inguinal adipocyte size are shown. Scale bar, 50 mm. * *p* < 0.05; compared with HFD-CT, # *p* < 0.05; compared with NCD-CT.

**Figure 2 molecules-23-03356-f002:**
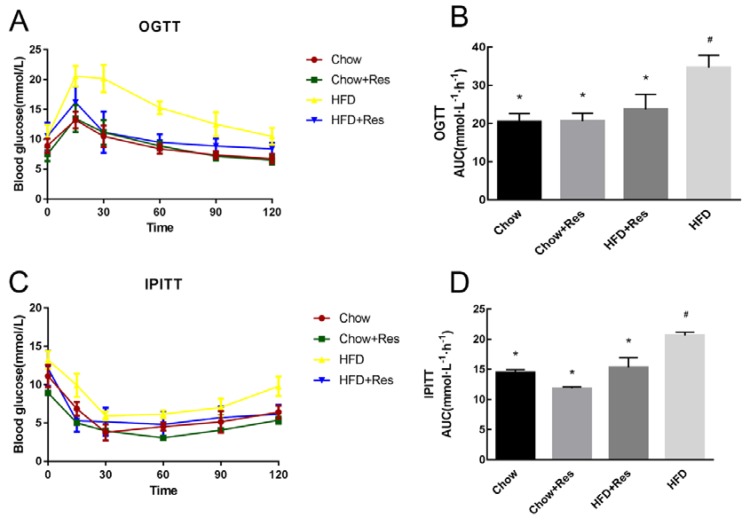
Resveratrol (RES) modulated glucose homeostasis. (**A**) Blood glucose levels and (**B**) Areas under the curve (AUC) during the oral glucose tolerance test (OGTT). (**C**) Blood glucose levels and (**D**) AUC during the intraperitoneal injection of insulin tolerance test (IPITT). * *p* < 0.05; compared with HFD-CT, # *p* < 0.05; compared with NCD-CT, (*n* = 5/group).

**Figure 3 molecules-23-03356-f003:**
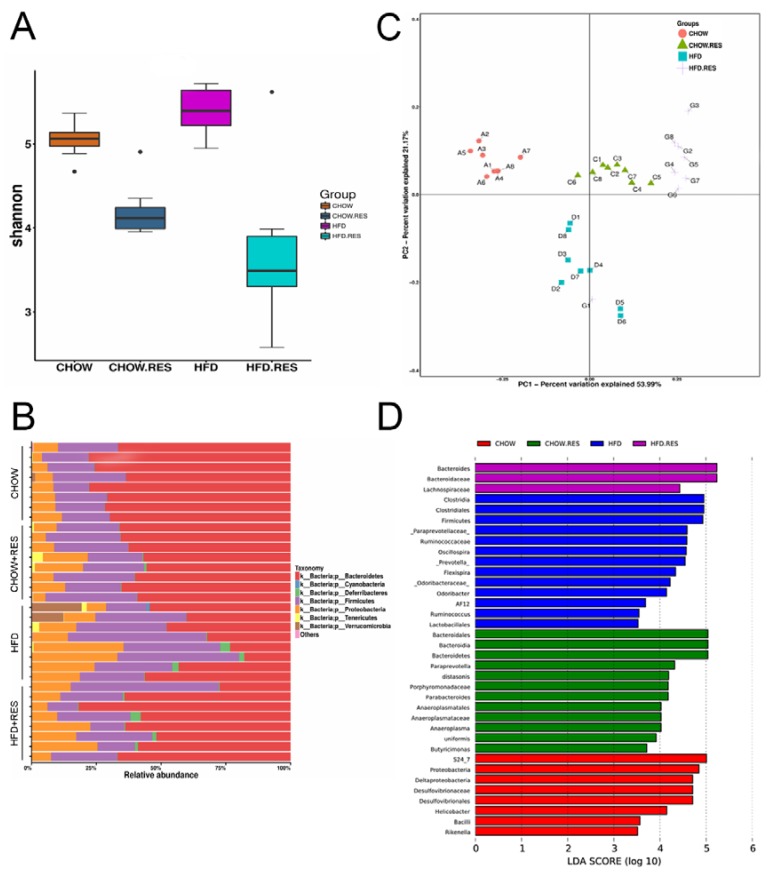
Resveratrol (RES) reversed high-fat diet (HFD)-induced dysbiosis. We performed a 454-pyrosequencing analysis of the bacterial 16S rRNA gene to assess the fecal microbiota composition of the four groups (*n* = 8 for each group). The Shannon index (**A**) of the fecal samples from different groups. (**B**) Bar charts showing the relative abundances (%) of different bacterial phyla in the different groups. (**C**) Bacterial communities were clustered using unweighted UniFrac distance-based principal coordinates analysis (PCoA). (**D**) LDA scores of the differentially abundant taxa. Taxa enriched in the gut microbiota from different diet and treatment groups are indicated by the LDA score (taxa with an LDA score > 3.5 and significance of α < 0.05 were determined by the Wilcoxon signed-rank test).

**Figure 4 molecules-23-03356-f004:**
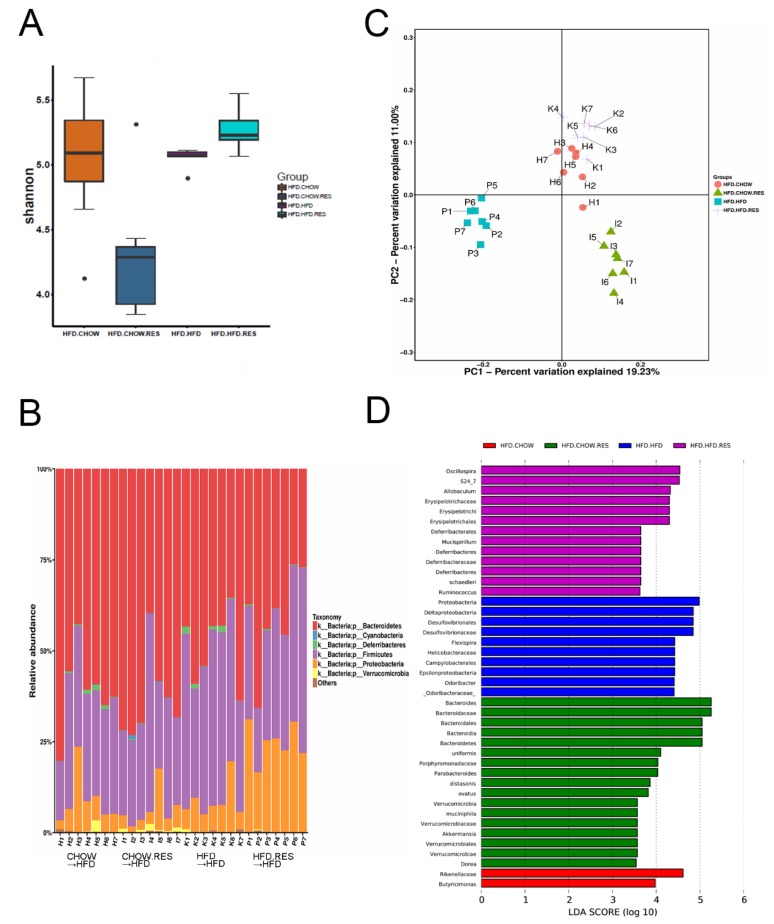
Resveratrol (RES) fecal transplants modulated gut microbiota composition. Fecal transplantation and relevant analyses were performed as described in the Methods section (*n* = 7 for each group). Shannon indices (**A**) of fecal samples from different groups. (**B**) Bar charts showing the relative abundances (%) of different bacterial phyla in the different groups. (**C**) Bacterial communities were clustered using unweighted UniFrac distance-based principal coordinates analysis (PCoA) (**D**) LDA scores of the differentially abundant taxa are shown. Taxa enriched in the gut microbiota from the different diet and treatment groups are indicated by the LDA score.

**Figure 5 molecules-23-03356-f005:**
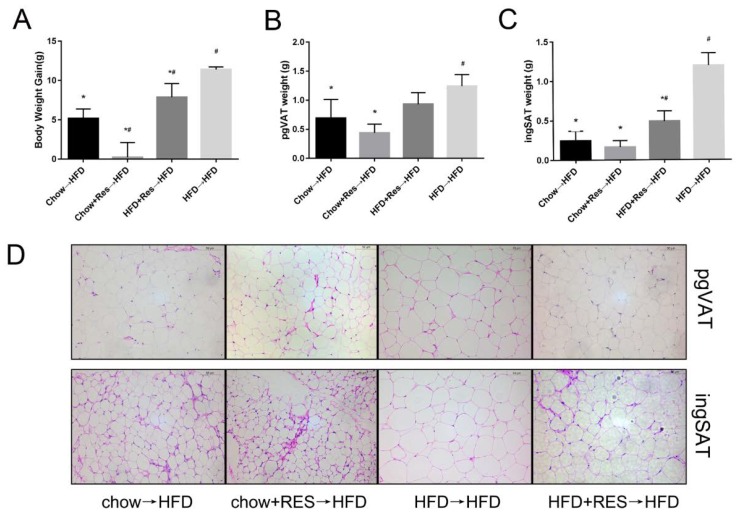
Resveratrol (RES) fecal transplants reduced high-fat diet (HFD)-induced obesity in mice. Effects of RES fecal transplantation on (**A**) body weight gain, (**B**) inguinal adipose tissue (ingSAT) weight and (**C**) perigonadal visceral adipose tissue (pgVAT) weight; (**D**) perigonadal visceral adipocyte and inguinal adipocyte sizes are shown. * *p* < 0.05; compared with HFD→HFD, # *p* < 0.05; compared with Chow→HFD.

**Figure 6 molecules-23-03356-f006:**
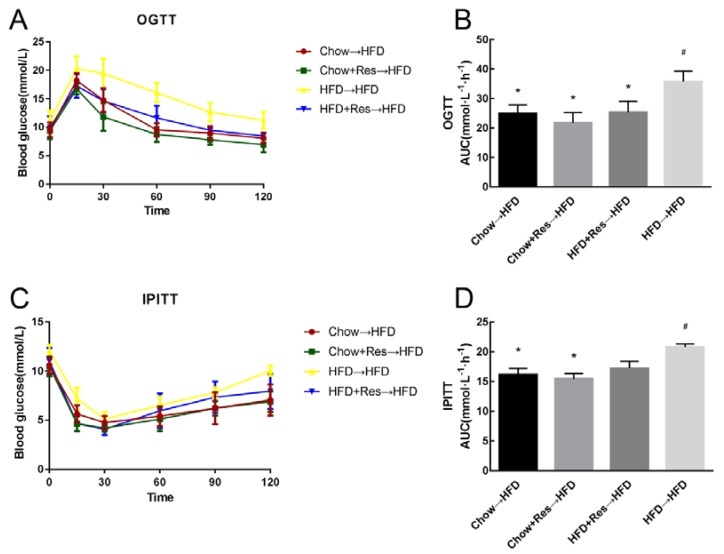
Resveratrol (RES) fecal transplants improved glucose homeostasis in high-fat diet (HFD)-fed mice. (**A**) Blood glucose levels and (**B**) Area under the curve (AUC) during the oral glucose tolerance test (OGTT). (**C**) Blood glucose levels and (**D**) AUCs during the intraperitoneal injection of insulin tolerance test (IPITT). * *p* < 0.05; compared with HFD→HFD, # *p* < 0.05; compared with Chow→HFD, (*n* = 5/group).

**Figure 7 molecules-23-03356-f007:**
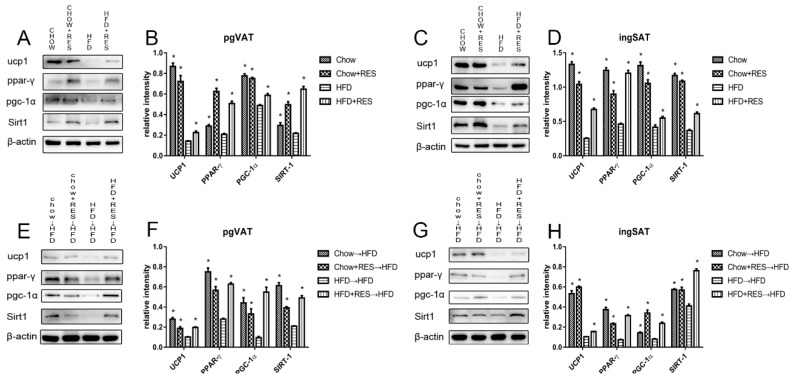
Resveratrol (RES) and fecal transplants from mice administered RES modulated the expression of proteins related to energy expenditure and white adipose tissue (WAT) browning. The effects of RES on the protein expression levels of UCP1, Sirt1, PGC-1α, UCP1, PPAR-γ and β-actin in (**A**) perigonadal visceral adipose tissue (pgVAT) and (**C**) inguinal adipose tissue (ingSAT) and densitometric analysis of these proteins (**B**,**D**). Representative adipose immunoblots for Sirt1, UCP1, PGC-1α, PPAR-γ and β-actin in (**E**) pgVAT and (**G**) ingSAT in recipient mice. Densitometric analysis of these proteins (**F**,**H**) in each group. * *p* < 0.05; compared with HFD-CT or HFD→HFD.

**Figure 8 molecules-23-03356-f008:**
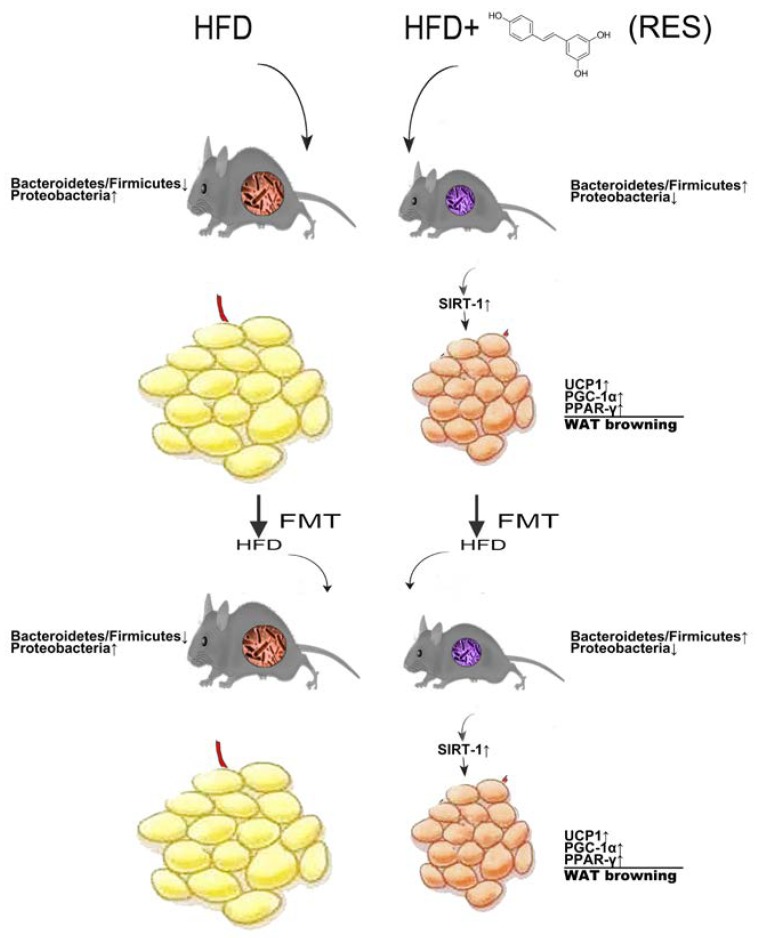
Proposed model for role of resveratrol (RES) in promoting white adipose tissue (WAT) browning in obese mice by remodeling the gut microbiota. RES treatment significantly inhibited increases in fat accumulation and promoted WAT browning in high-fat diet (HFD)-fed mice. Notably, RES supplementation significantly improved dysbiosis induced by HFD. Moreover, the weight-lowering and WAT-browning effects of RES in HFD-fed mice may be due to modulation of the gut microbiota.

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
