# Peer review of "Resveratrol-Induced White Adipose Tissue Browning in Obese Mice by Remodeling Fecal Microbiota"

_molecules, 2018, doi:10.3390/molecules23123356_

Round 1

Reviewer 1 Report

Comments and Suggestions for Author:

This study aimed to elucidate the potential anti-obesity effect of resveratrol (RSV) in animal model. Authors conclude that RSV is able to induce white adipose tissue (WAT) browning by microbiota remodeling.

The topic is of particular interest, the manuscript is well written, the experimental design is adequate, the results are well presented and the hypothesis that resveratrol could have potential anti-obesity effect is intriguing. There are however some major and minors concerns.

Major concern:

The Authors do not provide any cause-effect relationship by which the microbiota remodeling is the mechanism for the anti-obesity activity of RSV. This is the main issue of the present study, thus it should be more extensively discussed providing justifications in support of the hypothesis proposed.

Minor concerns:

·         Resveratrol should be abbreviated as RSV and not RES

·         Reference list – There is a lot of development in this field within the past 12 months. Authors should update their reference list with more recent references.

·         Since in last few years several Authors widely investigated further beneficial effects of RSV, driving the scientific research to elucidate novel properties of this polyphenols (despite those well-known and well-established), more information should be provided regarding chemical features of RSV as well as beneficial properties, including antioxidant, cardioprotective, neuroprotective, anticancer and antiviral activities (see ref: Baur JA Nat Rev Drug Discov 2006, 5, 493-506; Lopez MS Neurochem Int 2015, 89, 75-82; Lançon A Molecules 2016, 21, 304; Annunziata G Viruses 2018, 10, 473, doi: 10.3390/v10090473; Ardid-Ruiz A Nutrients 2018, 10, pii:E1757, doi:10.3390/nu10111757). This references should be included in the introduction

·         Abbreviations should be defined the first time they appear in the text. Abbreviations used in the abstract should be defined again if they are used in the main text. Abbreviations used in the results section should be defined.

·         Line #34: please write obesity with the lowercase letter

·         Line #99: please reformulate this sentence

·         Line #191: please reconsider the plural form

In this referee's opinion, with suggested revisions this manuscript is suitable for publication in Molecules journal.

Author Response

Dear Reviewer:

We were glad to have your comments about our article, therefore we discussed thoroughly for each of your suggestion or question. We have made the “point-by-point” responses as following and revised the manuscript according to reviewers’ comments.

Point 1: The Authors do not provide any cause-effect relationship by which the microbiota remodeling is the mechanism for the anti-obesity activity of RSV. This is the main issue of the present study, thus it should be more extensively discussed providing justifications in support of the hypothesis proposed.

Response 1: Thanks for your question. We extensively discussed the cause-effect relationship in the discussion. The results of our study suggested that RES could promote WAT browning in HFD mice. In order to determine whether this effect of RES-treated mice was caused by microbiota remodeling, we performed fecal microbiota transplantation (FMT) in HFD mice. Interestingly, improvements in RES-FMT mice we observed were similar to that in RES-treated mice. Our data appeared to exclude potential direct effects of circulating resveratrol on target tissues and proved that the anti-obesity effect is transmissible via FMT from RES-treated mice to HFD-fed mice. These abilities of RES are transferrable through colonization, supporting the theory that WAT browning is caused by altering gut microbiota composition.

Point 2: Resveratrol should be abbreviated as RSV and not RES.

Response 2: Thank you for reminding us. There are some different abbreviations for resveratrol in many related articles, such as RES, RSV, RV or RESV, and we chose “RES” as the abbreviation for resveratrol in this article (see ref: 1. Chen, S., Xiao, X., Feng, X., Li, W., Zhou, N., & Zheng, L., et al. (2012). Resveratrol induces sirt1-dependent apoptosis in 3t3-l1 preadipocytes by activating ampk and suppressing akt activity and survivin expression. Journal of Nutritional Biochemistry, 23(9), 1100.2. Qiao, Y., Sun, J., Xia, S., Tang, X., Shi, Y., & Le, G. (2014). Effects of resveratrol on gut microbiota and fat storage in a mouse model with high-fat-induced obesity. Food & Function, 5(6), 1241-1249.3. Wood, J. G. , Rogina, B. , Lavu, S. , Howitz, K. , Helfand, S. L. , & Tatar, M. , et al. (2004). Corrigendum: sirtuin activators mimic caloric restriction and delay ageing in metazoans. Nature, 430 (7000), 686-689.).

Point 3: Reference list – There is a lot of development in this field within the past 12 months. Authors should update their reference list with more recent references.

Response 3: Thanks for your advice. We updated the latest references in this field, but we still keep classical, far-reaching and famous studies in the reference list.

Point 4: Since in last few years several Authors widely investigated further beneficial effects of RSV, driving the scientific research to elucidate novel properties of this polyphenols (despite those well-known and well-established), more information should be provided regarding chemical features of RSV as well as beneficial properties, including antioxidant, cardioprotective, neuroprotective, anticancer and antiviral activities (see ref: Baur JA Nat Rev Drug Discov 2006, 5, 493-506; Lopez MS Neurochem Int 2015, 89, 75-82; Lançon A Molecules 2016, 21, 304; Annunziata G Viruses 2018, 10, 473, doi: 10.3390/v10090473; Ardid-Ruiz A Nutrients 2018, 10, pii:E1757, doi:10.3390/nu10111757). This references should be included in the introduction.

Response 4: Thanks for your advice. We searched further beneficial effects of resveratrol and updated related references in the introduction.

Point 5: Abbreviations should be defined the first time they appear in the text. Abbreviations used in the abstract should be defined again if they are used in the main text. Abbreviations used in the results section should be defined.

Response 5: Thank you for reminding us. We checked all abbreviations thoroughly in this article, and defined abbreviations when they first appear both in the abstract and in the main text.

Point 6: Line #34: please write obesity with the lowercase letter.

·         Line #99: please reformulate this sentence.

·         Line #191: please reconsider the plural form.

Response 6: Thanks for reminding us. We corrected the mistakes mentioned above in the text.

Thank you very much for your nice suggestion and assist during the revision.

Thank you and best regards.

Yours sincerely,

Corresponding author: Xiang Feng Assoc. Prof, M.D., Ph.D.

Dept. of Nutrition, School of Public Health, Sun Yat-sen University

Reviewer 2 Report

This paper is very interesting and contains high-value results. Whole work has been well designed. All sections are clearly described. I found only a few minor issues which should be corrected before acceptation of this paper.

Language is very poor, need to be corrected by a native speaker

Line 64-66 the sentence needs to be a rewrite

Lines 38, 59, 71, 101, 102, 107, 274 editorial bug

Figure 3 is very hard to read I suggest to try to do two separate figures from it with a better scale. Similar problem I see in figure 4

Lack of antibiotic origin description

Line 269 Sigma Aldrich is not China; you should give the origin of the company, not a local reseller

Figure 8 in a very interesting way summarising this paper, while its resolution is very low, Authors need to improve this issue.

Author Response

Dear Reviewer:

We were glad to have your comments about our article, therefore we discussed thoroughly for each of your suggestion or question. We have made the “point-by-point” responses as following and revised the manuscript according to reviewers’ comments.

Point 1: Language is very poor, need to be corrected by a native speaker.

Response 1: Thanks for your advice. We used a professional English editing service to improve the language, and all corrected place were tracked.

Point 2: Line 64-66 the sentence needs to be a rewrite.

Response 2: Thanks for your advice. We rewrote this sentence to make our intended meaning more clearly.

Point 3: Lines 38, 59, 71, 101, 102, 107, 274 editorial bug.

Response 3: Thank you for reminding us. We corrected these editorial bugs.

Point 4: Figure 3 is very hard to read I suggest to try to do two separate figures from it with a better scale. Similar problem I see in figure 4.

Response 4: Thanks for your advice. Since the Fig. 3D and Fig. 4D were too long, we adjusted LDA score and redid the Fig. 3D and Fig. 4D to make figure 3 and 4 a better scale.

Point 5: Lack of antibiotic origin description

Response 5: Thank you for reminding us. We added descriptions of antibiotic.

Point 6: Line 269 Sigma Aldrich is not China; you should give the origin of the company, not a local reseller

Response 6: Thank you for reminding us. We corrected the origin of Sigma Aldrich in Materials and Methods.

Point 7: Figure 8 in a very interesting way summarizing this paper, while its resolution is very low, Authors need to improve this issue.

Response 7: Thank you for reminding us. We revised figure 8 and updated higher resolution version of this figure.

Thank you very much for your nice suggestion and assist during the revision.

Thank you and best regards.

Yours sincerely,

Corresponding author: Xiang Feng Assoc. Prof, M.D., Ph.D.

Dept. of Nutrition, School of Public Health, Sun Yat-sen University

Round 2

Reviewer 1 Report

The authors have adequately fulfilled this reviewer's suggestions which were proposed in the first revision of the manuscript.